# The Hidden Secrets of the Dental Calculus: Calibration of a Mass Spectrometry Protocol for Dental Calculus Protein Analysis

**DOI:** 10.3390/ijms232214387

**Published:** 2022-11-19

**Authors:** Omer Bender, Dana Megreli, Talia Gavish, Noa Meyrom, Neta Zamir, Hila May, Rachel Sarig, Daniel Z. Bar

**Affiliations:** 1Department of Oral Biology, The Goldschleger School of Dental Medicine, Faculty of Medicine, Tel Aviv University, Tel Aviv-Yafo 69978, Israel; 2Dan David Center for Human Evolution and Biohistory Research, Sackler Faculty of Medicine, Tel Aviv University, Tel Aviv-Yafo 69978, Israel; 3Department of Anatomy and Anthropology, Sackler Faculty of Medicine, Tel Aviv University, Tel Aviv-Yafo 69978, Israel

**Keywords:** dental calculus, mass-spectrometry, modern humans, human evolution

## Abstract

Dental calculus is a solid deposit that forms and accumulates on the tooth surface, entrapping oral microorganisms, biomolecules, and other micro-debris found in the oral cavity. A mass spectrometry analysis of its protein content opens a vista into the subject’s diet, oral flora, and even some aspects of health, thus providing new insight and expanding our knowledge of archaic cultures. Multiple experimental protocols have been proposed for the optimal extraction of proteins from dental calculus. Herein, we compared various experimental conditions in order to calibrate and validate a protocol for protein extraction. Our results show that a high concentration of acetic acid followed by mechanical crushing and sonication provided the highest protein yield, while acetone precipitation enabled the identification of more distinct proteins. We validated this protocol using archeological samples, identifying human and microbial proteins in specimens from the eighth and seventeenth centuries (approximately 250–1300 years ago). These findings demonstrate that the developed protocol is useful for studying excavated archaeological samples and that it might be utilized to explore the biohistory, dietary habits, and microbiome of archaic populations.

## 1. Introduction

Dental calculus, also called tooth tartar, is a solid deposit that forms and accumulates on the tooth surfaces and creates a firm bond with the teeth that can only be removed by mechanical means. Dental calculus originates from a mineralized oral bacterial plaque that naturally forms on tooth surfaces where there is a constant supply of saliva. The mineralization process is caused by the deposition of calcium phosphate crystals from the saliva into the bacterial plaque [1]. Since calculus formation is related to saliva sources, it is commonly seen over the buccal surfaces of maxillary molars and lingual surfaces of mandibular anterior teeth, where the salivary ducts open into the oral cavity [2].

The calcification process of dental plaque is dynamic: it begins with the adsorption of salivary proteins on the enamel surfaces, followed by adherence of a structurally organized biofilm, which eventually calcifies in several phases of mineralization. During the process, the precipitation of mineral ions (calcium and phosphate) from the saliva and gingival crevicular fluid is settled in the bacterial biofilm, together with additional oral microorganisms, biomolecules, and other micro-debris sourced directly from the human mouth [3].

The calculus matrix consists mainly of inorganic salts and mineral ions deriving from saliva, (e.g., hydroxyapatite, calcium, whitlockite, and phosphorus). It also consists of organic compounds, including microfossils, (e.g., starch granules, phytoliths, and plant metabolites) and biomolecules, (e.g., proteins, lipids, and nucleic acids) derived from consumed foods, the oral microbiome, or the host [3,4].

As dental calculus captures and preserves microparticles throughout an individual’s lifetime, it may contain valuable information. Due to its densely mineralized nature, calculus can serve as a long-term abundant reservoir of ancient proteins that can last for extended time periods and preserve numerous biomolecules containing valuable information. Therefore, archaeological dental calculus serves as a fossilized record of the oral environment. A single sample of calculus contains numerous molecules that can provide information about the individual’s genome [5,6], oral microbiome and health [7], diet [8,9], and even occupational activities [10].

Changes in prehistoric populations’ lifestyles are determined by diet, culture, and habits [11]. Therefore, these parameters can be reflected by the composition of dental calculus deposits [12]. Accordingly, an analysis of dental calculus offers a unique opportunity to reveal the everyday function of individuals and gain insight into the lifestyle and dietary habits of prehistoric populations. In addition to other dietary molecules, the dental calculus specifically preserves food-related proteins and, by a proper extraction method, we can detect proteomic evidence of a variety of food ingredients [13].

Moreover, as the human mouth is a habitat ecosystem for a variety of microorganisms, the dental calculus preserves proteins associated with the oral microbiota and might contain disease-associated microorganisms, including viruses and fungi [14,15]. Archaeological dental calculus is a rich source of ancient proteins derived from the microbiome and thus serves as a tool for palaeomicrobiology (the study of microorganisms from ancient sources) and has great potential to elucidate the evolution of the human microbiome. This knowledge might shed light on human-associated microbes in a variety of acute and chronic diseases and provide a valuable and vital source of information for assessing individual health [14,16].

### 1.1. Mass Spectrometry (MS) for Quantitative Proteomics Analyses

MS is a powerful method for hypothesis-free and hypothesis-driven identification of proteins. By applying this method to mineral deposits, we can detect the presence of protein molecules in the sample. The MS-based proteomics workflow requires an extraction procedure for the existing proteins, which are then digested into peptides. The peptide mixture is eventually analyzed by MS [8,17]. The timeframe in which these data can be retrieved varies and depends, among others, on the preservation conditions of the archaeological specimens in the excavated sites.

### 1.2. Limitations

In recent years, dental calculus has become the subject of an increasing number of investigations that concern different research fields in biology [5,6,7,8,9,12,14,17,18,19]. However, the quantity of archaeological dental calculus available per individual is generally low, sometimes only tens of milligrams per sample. This is partially due to preservation protocols that historically included cleaning the archaeological specimens and removal of any residual calculus, or in cases where only a few isolated teeth were recovered from the sites. This factor limits the number of analyses that can be performed for the extraction of biomolecules from different sources [5]. According to these limitations, it is evident that the amount of dental calculus has become the main limiting factor for obtaining data. Thus, a reliable, reproducible, and effective protein extraction protocol allowing to obtain considerable information from archaeological samples is highly required.

Recent studies suggested various extraction protocols to evaluate archaic dental calculus using metaproteomics [17,19] and yet, there is still a debate as to what is the most effective protocol for such purposes. Herein, we provide an optimized protocol for protein content analysis from dental calculus samples. We compared various conditions for decalcification, mechanical crushing, sonication, protein precipitation, and MS gradient times. These resulted in a single optimized protocol (Appendix A) that can be used to deduce individual dietary practices and inform us on oral health and oral microbiome composition in a specific population.

## 2. Results

### 2.1. Decalcification Optimization

To enable efficient extraction of proteins from the sample, the dental calculus first needed to be decalcified. This is traditionally performed by a combination of acid treatment with mechanical disintegration of the sample. Here we used different concentrations of acetic acid (AA, 15%, 20%, and 25%). Samples were decalcified in the AA followed by manual crushing of the calculus using a micro-pestle. Protein concentration (Table 1) released after the decalcification and mechanical crushing was measured based on peptide backbone and aromatic side absorption (205 nm) using a nanodrop spectrophotometer.

Next, we tested the effect of sonication on protein release. Spectrophotometric measurements confirmed no protein loss from the decalcification and mechanical crushing.

Protein yield increased with acid concentration and required mechanical crushing (Figure 1). Sonication had a positive but very mild effect on protein yield. The greatest amount of protein yield was achieved using 25% AA followed by mechanical crushing of the calculus and sonication. However, protein elution was not complete, and repetitive elution may be advised for limited samples.

### 2.2. Protein Precipitation

Prior to protein digestion and mass spectrometry analysis, the protein content should be purified. We have tested which protein precipitation method results in higher protein yield (Table 2).

Without protein purification, no protein digestion using trypsin was detected (Figure 2). Both methanol-chloroform and acetone precipitation methods facilitated sample purification to allow the full cleavage of the proteins into peptides (Figure 2). However, the protein yield retrieved from acetone precipitation was lower compared to methanol-chloroform precipitation. This result was more noticeable on the gel (Figure 2), but consistent with the measurements in Table 2.

### 2.3. Gradient Time Optimization

Protein MS typically relies on peptide elution along an organic solvent gradient. Here, we compared the 1 h vs. 2 h time gradients of protein samples following two different precipitation methods: methanol-chloroform and acetone (Figure 3). To test these experimental conditions, we limited the bioinformatic search of the results to two sources, human proteins and *Escherichia coli* proteins. Some human proteins are known to be trapped in dental calculus and thus are expected to be found in all samples. By contrast, while *E. Coli* is fairly common in the digestive tract, it is present only in small numbers in the oral cavity (approximately 5/100,000 Operational Taxonomic Units by 16S data generated in our lab). Thus, a successful identification of *E. Coli* proteins will suggest the ability to identify proteins from less common sources.

After setting the minimal peptide count to three, methanol-chloroform precipitation resulted in twelve identified *E. Coli* proteins for both the 1 and 2 h run gradients (Figure 3). By contrast, after acetone precipitation, 13 proteins were identified in the 2 h run, compared to 11 proteins after the 1 h run (Figure 3). With no dramatic differences at the protein level, we analyzed these data at the peptide level. When examining the distribution of the peptide count following methanol-chloroform precipitation, only two proteins were identified with the same number of peptides in both runs, five proteins had more peptides in the 2 h run, one protein had more peptides in the 1 h run, two proteins were present only in the 2 h run, and four proteins were present only in the 1 h run (Figure 4a–d). After acetone precipitation, three proteins were identified with the same number of peptides in both runs, four proteins had more peptides in the 2 h run, two proteins had more peptides in the 1 h run, four proteins were present only in the 2 h run and two proteins were present only in the 1 h run (Figure 4a–d). Overall, these data suggest that a 2 h gradient typically identifies slightly more proteins and peptides.

Human proteins are known to be abundant in the saliva and gingival crevicular fluid trapped within the dental calculus [7]. Thus, calibrating their analysis will help to identify proteins of lower abundance. In this regard, a 1 h following methanol-chloroform precipitation identified 12 proteins, compared to 14 proteins identified in the 2 h run (Figure 3). When comparing proteins identified after acetone precipitation, 23 proteins were identified in the 2 h run compared to 19 in the 1 h run (Figure 3). In agreement with the *E. coli* data, a 2 h gradient was likely to identify more proteins, at the expense of longer run times.

### 2.4. Archaeological Dental Calculus Analysis

To demonstrate the applicability of the established protocol to identify proteins in archaeological samples, we performed proof-of-principle experiments on two samples that were retrieved from archaeological excavations, one was dated to the seventeenth century and the other was dated to the eighth century. In the Early Arab sample, we identified sixteen human proteins and six *E. Coli* proteins (Figure 5a). In dental calculus from the Late Arab specimen, we identified nineteen human proteins and nine *E. Coli* proteins (Figure 5b,c).

Another aspect of proteomics from archaeological calculus is to ascertain the dietary regimen of ancient populations. We searched the generated data for proteins of the Bovidae family. This family includes three domesticated mammals commonly used as food, as well as multiple members not found in the region. From the eighth century dental calculus, we have identified seven proteins: one is from goats, three are from the bos genus, including modern cattle (bos genus), and three are from sheep. From the seventeenth century dental calculus, we have identified eight proteins, four of them are from sheep, three are from modern cattle, and two are from goats (Figure 6). 

## 3. Discussion

Here, we present a protocol for proteomic analysis of modern and historic dental calculus samples. By carefully calibrating protein extraction and digestion, we were able to identify human and common bacterial proteins from modern and ancient samples. The data files generated can be used to search databases of commonly consumed plants and animals, as well as against the oral microbiome.

A broad range of protocols to decalcify and prepare both fresh and ancient calculus for further evaluation has been previously presented. Calcification methods vary from the use of EDTA [7,19] to various acids at different concentrations, including nitric acid [20], HCl [14,21,22], and acetic acid [8,17], with [20,21] or without manual grinding, which also varies in duration and temperature of calculus denaturation. Acetic acid (AA) was selected as the preferred agent among all the acids that can be utilized during the decalcification phase. Much research primarily employs this acid, since it effectively decalcifies dental calculus. However, it should be mentioned that other decalcifying substances might be effective as well and have not been tested [8,17]. We assessed different concentrations of acetic acid and the decalcification time required to extract proteins from the calcified dental calculus. Our results suggest that using a higher concentration of acid followed by mechanical crushing and ultrasonic disturbance is the optimal way to extract proteins from calcified calculus. 

While methanol-chloroform precipitation yielded more protein, we found that acetone precipitation provided a higher number of identified proteins and peptides. This may be the result of better retention of a large number of proteins.

The sample size is a common limiting factor in ancient calculus. We have shown that using proteomics analysis, proteins can be identified from a relatively small amount of calculus (~10–15 mg). These amounts can be further down-scaled, in order to allow proteome analysis of ancient calculus samples, but this will likely result in a decrease in the number of proteins identified.

Ample valuable data regarding the oral microbiome, ingested food, and oral and physical health can be gathered from the proteins extracted from dental calculus. Our results establish a valuable and effective protocol for gaining a considerable amount of data from a limited source of calculus. As proof of principle, we correctly identified three members of the Bovidae family used as a food source. Dental calculus proteomics can be used in combination with other anthropological and archaeological research approaches aimed to increase our understanding of the biohistory of archaic populations.

## 4. Material and Methods

For MS calibration, a minimum amount of 10–15 mg of dental calculus was considered necessary to obtain significant results in each experiment. For accurate comparison and optimal protocol establishment, all dental calculus used was collected from six patients at the clinic following signed and informed consent. Medical confidentiality was guaranteed regarding the identity of patients. The study was approved by the ethical committee of Tel Aviv University (Number 0005110-2). All research was performed in accordance with relevant guidelines and regulations. The fresh dental calculus was obtained using a sterile dental curette and stored in sterile 2.0 mL tubes until further analysis.

The established protocol was further applied to the archaeological samples retrieved from the anthropological collection at the Dan David Center for Human Evolution and Biohistory Research, Sackler Faculty of Medicine, Tel Aviv University. Two samples were obtained from excavated sites dated to Late Arab (the seventeenth century) and Early Arab (eighth century) periods.

### 4.1. Sample Preparation

In order to analyze the proteins captured in human calculus, the dental calculus was decalcified with 15%/20%/25% acetic acid overnight and the pellet was treated with/without mechanical crushing followed by sonication. Then, prior to MS analysis, samples were cleaned with different precipitation methods to remove detergents followed by a 1 or 2 h run time for MS analysis (Figure 7). All the steps are detailed below.

### 4.2. Decalcification

Upon usage, the fresh dental calculus samples retrieved from patients were washed three times with a PBS buffer for 10 min. After each wash cycle, the calculus was spun down at 10,000 RCF. The calculus samples were then demineralized in 15%, 20%, or 25% acetic acid (Biolab, Jerusalem, Israel) overnight on a 3D orbital shaker. Following demineralization, samples were grounded mechanically using an Eppendorf micro pastel (Merck, Rahway, NJ, USA). Samples were washed four times with PBS for 10 min. After each cycle of washing, samples were spun down at 10,000 RCF. Calculus samples were resuspended in a lysis buffer (2M Guanidine Hydrochloride (Merck, Rahway, NJ, USA), 10 mM Chloroacetamide (Bar Naor-Israel, Petah Tikwa, Israel), and 5 mM HEPES (Merck, Rahway, NJ, USA), pH 7.5), followed by sonication for 25 min at 40%. Samples were spun down at 21,000 RCF for 30 min at 4 °C.

### 4.3. Residual Protein Loss during Decalcification

Protein concentrations were measured using the A205 preprogrammed direct absorbance application of Nanodrop One C (Thermo Fisher Scientific, Waltham, MA, USA). Measurements were performed following acetic acid incubation and after the mechanical crushing to exclude the possibility of protein loss in washes.

### 4.4. Protein Precipitation

Following optional sonication, each sample was divided into two equal aliquots which were further processed using two different precipitation methods.

#### 4.4.1. Methanol-Chloroform

Methanol at four times (600 μL) the sample volume (150 μL) was added and vortexed thoroughly. Next, chloroform at the original sample volume was added and vortexed. Next, 400 μL of double distilled water (DDW) was added and the sample was vortexed thoroughly until the mixture became cloudy with precipitation. The sample was centrifuged at 14,000 RCF for 1 min. Three layers were apparent: a top aqueous layer and a circular flake of protein. The top layer was removed without disturbing the protein at the interface. Methanol at four times of sample volume was added and vortexed. The sample was centrifuged at 20,000 RCF for 5 min, and excess methanol was discarded. Then, the sample was placed under a vacuum and resuspended with 100 uL of a 100 mM HEPES buffer (Merck, Rahway, NJ, USA).

#### 4.4.2. Acetone

Cold (−20 °C) acetone at four times of sample volume was added and vortexed thoroughly. The sample was then incubated at −80 °C for 60 min and centrifuged at 15,000 RCF for 10 min at 4 °C. The supernatant was discarded. Then, the sample was placed under a vacuum for complete removal of the acetone and resuspended with 100 uL of a 100 mM HEPES buffer (Merck, Rahway, NJ, USA).

The amount of protein extracted from the calculus was determined using a Bradford (Merck, Rahway, NJ, USA) assay and measured with a nanodrop (Nanodrop one C, ThermoFisher, Waltham, MA, USA) at absorbance 595 nm with a baseline correction of 750 nm.

### 4.5. Protein Processing

Following protein precipitation, samples underwent reduction by adding Dithiothreitol (Merck, Rahway, NJ, USA) to a final concentration of 10 mM for 60 min at 55 °C, followed by alkylation by adding Iodoacetamide (Merck, Rahway, NJ, USA) to a final concentration of 18.75 mM for 30 min at room temperature. Subsequently, samples were digested overnight at 37 °C with a 10 nG MS Grade Trypsin (Promega, Madison, WI, USA). There was 1.5 μG of protein subjected to trypsinization. The enzymatic reaction was quenched with 1 μL of 0.1% Trifluoroacetic acid (TFA) (ThermoFisher, Waltham, MA, USA).

### 4.6. Solid Phase Extraction Stage Tips for Detergent Removal

Prior to mass spectrometry, samples were cleaned with a solid phase extraction method using two layers of C18 Empore^TM^ SPE Disks (Merck, Rahway, NJ, USA) followed by two layers of SCX strong cation exchange Empore^TM^ SPE Disks (Merck, Rahway, NJ, USA). Samples were then lyophilized.

### 4.7. MS Analysis

An MS analysis was performed at the National Proteomics Unit at the Technion, Israel. The peptides were resolved by reverse-phase chromatography on 0.075 X 300-mm fused silica capillaries packed with Reprosil reversed phase material C18 (Dr Maisch GmbH, Ammerbuch, Germany). The peptides were eluted using either a 60 linear minutes gradient or a 120 min linear gradient of 5% to 28%, a 15 min gradient of 28% to 95%, and 15 min at 95% Acetonitrile with 0.1% formic acid in water at flow rates of 0.15 μL/min. An MS analysis was performed by a Q Exactive HF mass spectrometer (Thermo Fisher Scientific, Waltham, MA, USA) in a positive mode (*m/z* 300–1800, resolution 120,000 for MS1 and 15,000 for MS2) using a repetitively full MS scan followed by high collision dissociation (HCD, at 27 normalized collision energy) of the 20 most dominant ions (≥1 charges) selected from the first MS scan. The AGC settings were 3 × 10^6^ for the full MS and 1 × 10^5^ for the MS/MS scans. The intensity threshold for triggering MS/MS analysis was 1.3 × 10^5^. A dynamic exclusion list was enabled with an exclusion duration of 20 s.

### 4.8. Silver Stain

Fifteen percent SDS-PAGE gels were prepared in advance. Gels were made with two parts: running gel and stacking gel. Running gel was made with 1.2 mL DDW, 5 mL 30% Acrylamide 29:1 (Bar-Naor, Petah Tikwa, Israel), 1.25 mL 1.5 M Tris-HCl pH 8.8, 50 uL 10% Sodium Dodecyl Sulfate (SDS, Thermo-Fisher, Waltham, MA, USA), 25 uL 10% Ammonium Persulfate (APS, Merck, Rahway, NJ, USA), 2.5 uL TEMED (Bar-Naor, Petah Tikwa, Israel). Stacking gel was made with 2.3 mL DDW, 500 uL 30% Acrylamide 29:1, 950 uL 0.5 M Tris-HCl pH 8.8, 37.5 uL 10% SDS, 18.75 uL 10% APS, 3.75 uL TEMED. 

Samples were loaded with an X4 Laemli Sample Buffer (LSB, #1610747, Bio -Rad, Hercules, California, USA) and put in a preheated heating block set to 95 °C for 5 min. Samples were then separated on the 15% SDS-PAGE gels with 120 mV constant voltage. Gels were then stained using a MS-compatible silverquest staining kit (Thermo-Fisher LC6070, Thermo-Fisher, Waltham, MA, USA) according to manufacturer instructions.

### 4.9. Archeological Dental Calculus Samples

Following the calibration of the protocol, the procedure was applied to the dental calculus samples retrieved from the archeological sites. Each sample was treated as stated in our protocol.

### 4.10. Bioinformatic Analysis

To assess the effect of the precipitation method and gradient time on the obtained data, we searched the data for peptides and proteins from two sources expected to be present in our sample: proteins of human origin and *E. Coli.* Mass-spec raw data files were run on the MaxQuant V2.1.0.0 software (Max Planck Institute) to identify peptides and proteins. Protein minimal peptide count per protein was set to three, with an FDR set to 10% per peptide. The following protein FASTA files were used for protein identification:Homo Sapiens (Human)—proteome ID UP000005640.Escherichia Coli—proteome ID UP000036496.Bovidae—proteome ID UP000009895.

## Figures and Tables

**Figure 1 ijms-23-14387-f001:**
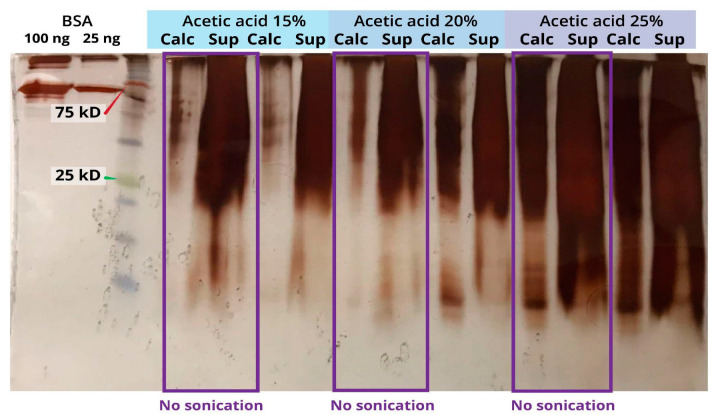
Silver-staining estimate of protein yield under different conditions. Calc.—calculus. Sup.—supernatant.

**Figure 2 ijms-23-14387-f002:**
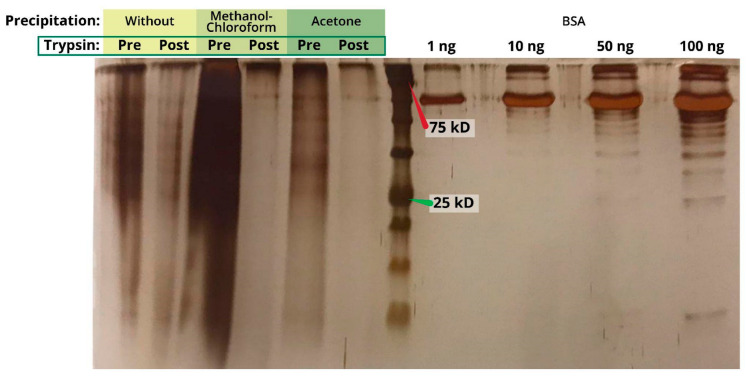
Silver-staining of proteins separated on an SDS-PAGE gel following either of two methods of precipitation. Methanol-chloroform and acetone precipitation products before and after trypsinization of the 25% AA protein samples are shown.

**Figure 3 ijms-23-14387-f003:**
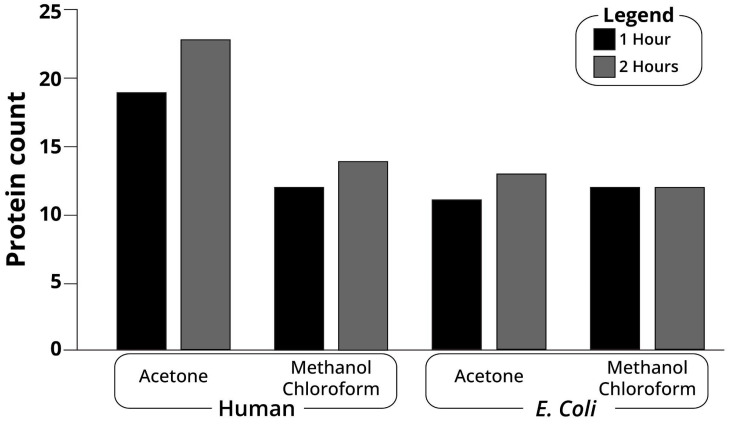
Comparison of the MS samples running time. A. 1 vs. 2 h run of *E. Coli* and human proteins after methanol-chloroform and acetone precipitation.

**Figure 4 ijms-23-14387-f004:**
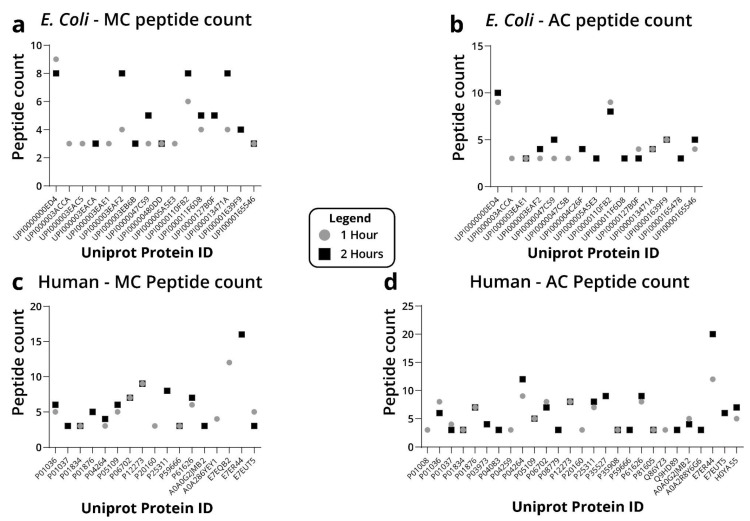
Number of peptides identified for each protein identified by 1 and 2 h MS run and their Uniprot ID. (**a**) Peptide count of *E. Coli* proteins after methanol-chloroform precipitation. (**b**) Peptide count of *E. Coli* proteins after acetone precipitation. (**c**) Peptide count of human proteins after methanol-chloroform precipitation. (**d**) Peptide count of human proteins after acetone precipitation.

**Figure 5 ijms-23-14387-f005:**
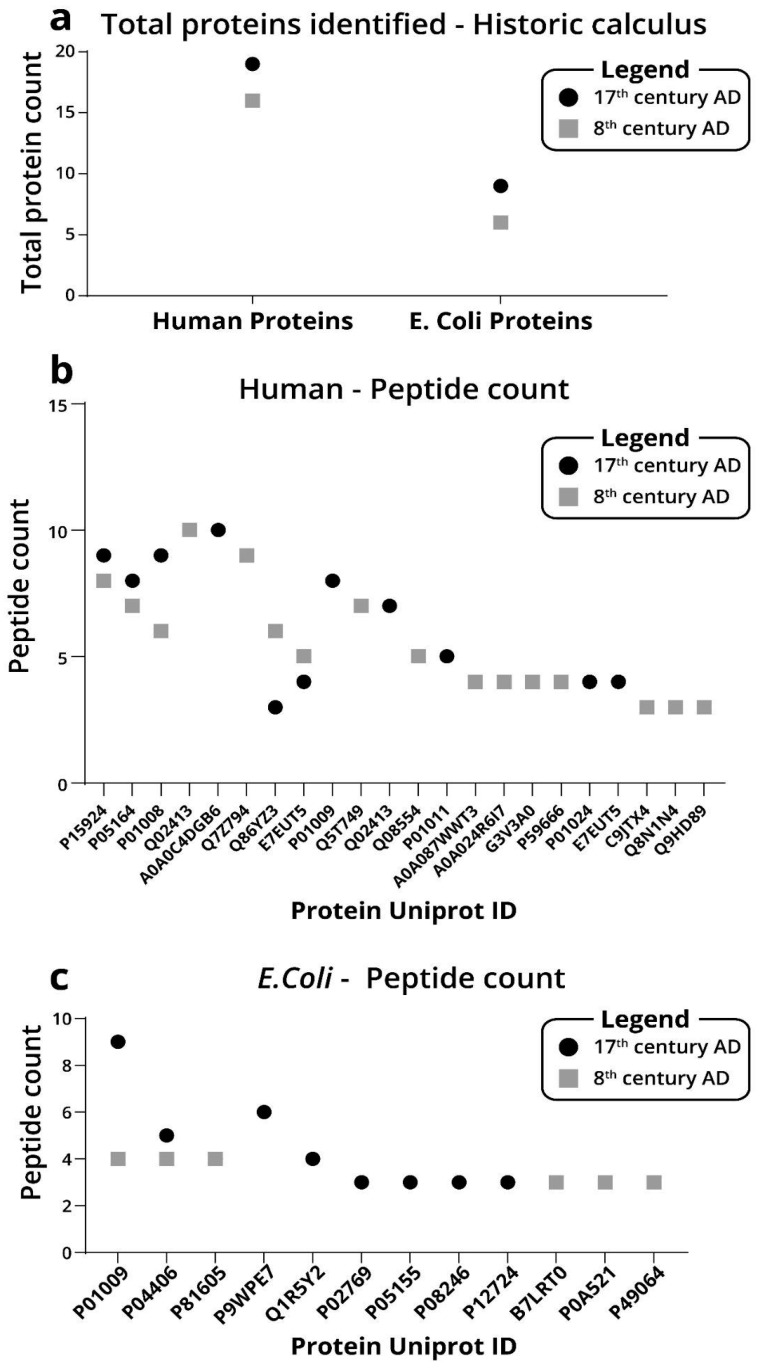
Number of peptides identified for each protein identified by 2 h MS run. (**a**) protein count of human proteins from historic dental calculus. (**b**) Peptide count of human proteins from historic dental calculus. (**c**) Peptide count of *E. Coli* proteins from historic dental calculus.

**Figure 6 ijms-23-14387-f006:**
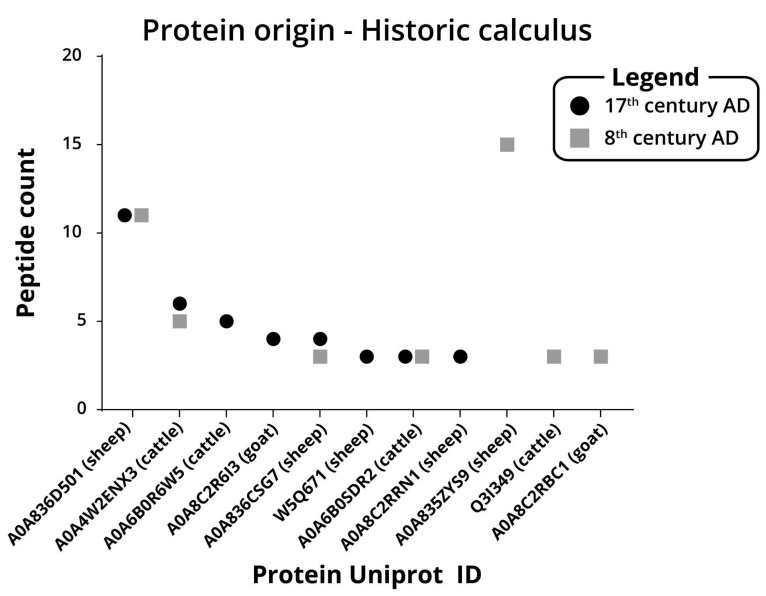
Number of peptides identified for each protein identified from Bovidae family proteins. Peptide count of Bovidae family origin proteins from historic dental calculus. Parentheses indicated the protein with the most peptides after the maxquant search.

**Figure 7 ijms-23-14387-f007:**
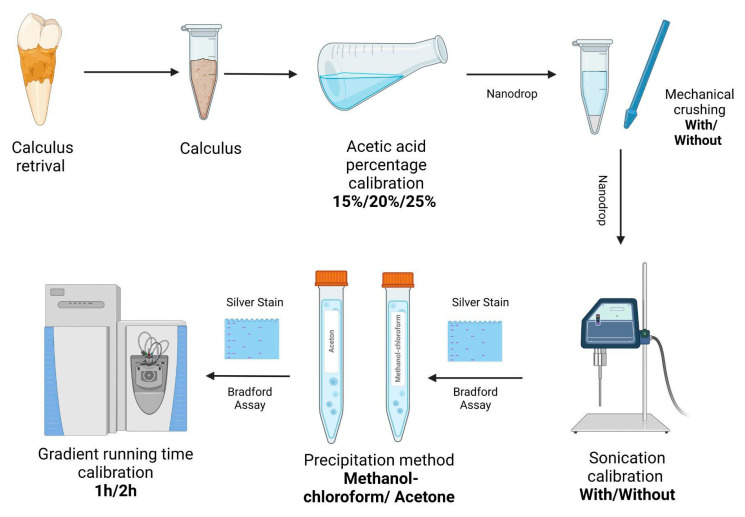
Protocol schematic overview.

**Table 1 ijms-23-14387-t001:** Protein concentration after demineralizing the calculus with or without mechanical crushing. Measured by NanodropOne C at 205 nm with the 31-correction method.

Acetic Acid Concentration	Protein Concentration mg/mL before Mechanical Crushing	Protein Concentration mg/mL after Mechanical Crushing
20%	0.13	0.23
25%	0.23	0.27

**Table 2 ijms-23-14387-t002:** Protein concentration before and after acetone or methanol-chloroform precipitation.

Acetic Acid Concentration	Protein Concentration μg/mL before Precipitation	Protein Concentration μg/mL after Acetone Precipitation	Protein Concentration μg/mL after Methanol—Chloroform Precipitation
20%	134	130	126
25%	154	138	141

## Data Availability

Data generated are available at MassIVE MSV000090092 ftp://massive.ucsd.edu/MSV000090092/, https://massive.ucsd.edu/ProteoSAFe/dataset.jsp?task=d10befd7a5ac45c385362f32ed936e4d, accessed on 16 November 2022.

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
