# Peer review of "The Hidden Secrets of the Dental Calculus: Calibration of a Mass Spectrometry Protocol for Dental Calculus Protein Analysis"

_ijms, 2022, doi:10.3390/ijms232214387_

Round 1

Reviewer 1 Report

The authors present a practical and straightforward sample preparation approach for extracting protein from the dental calculus. The biological and historical relevance of these samples is just starting to be explored and the development of methods that can extend to older samples is of value. The author rationale and selection of proof-of-concept samples is logical. Towever, the authors are mainly presenting their results as a technical resource, and in this respect, several points have not been addressed in the current manuscript. Specifically, the authors discuss many different extraction solutions, but only evaluated one, acetic acid.  It may be that the modern calculus samples are limited, but in this case, the authors should justify why acetic acid was selection over other possibilities. Additionally, the justification for specific methodological sample preparation steps and key details for reproducing the method are lacking (see detailed points below).  Finally, despite a proof-of-concept manuscript, the potential significance of the findings could be discussed.

Major point related to significance

While the authors did not perform a study with multiple biological replicates, perhaps they could comment on selected proteins of interest detected in historic samples that were not present in modern calculus samples and  discuss any implications from a paleobiology perspective.

Major points related to methodology:

-The authors performed and compared protein clean-up steps to facilitate efficient protein digestion with trypsin. However, their choice to reduce and alkylate the sampe after clean-up was unexpected.  It appears the authors included an alkylating agent, chloroacetamide, in the lysis buffer, but not a reducing agent, such as TCEP, which can be used at the same time as chloroacetamide.  Did the authors attempt to reduce and alkylate during the lysis step as this is most logical point?

-The amount of protein yield extracted from the calculus or after precipitation should be reported. Additionally, how much of that protein was subjected to trypsin digestion should be included.

-Performing reduction and alkylation as stated in the methods would result in excess iodoacetamide that is incubated at 37C with trypsin, and potentially generating unwanted protein modifications and reducing protein digestion effectiveness.  To provide evidence for or against this issue, a missed cleavage analysis of the peptide identified would be useful.

-Additionally, given the relatively harsh extraction conditions, the authors should also have performed bioinformatics analysis to search for artifacts of sample preparation or semi- and non-tryptic digestion products. This is particularly relevant for archaeological samples where a wider variety of products and oxidative artefacts could be present. I would suggest the authors explore the use of FragPipe software, https://fragpipe.nesvilab.org/, which is optimized to search for variable modifications and unexpected modifications. The authors may even find that after carefully optimizing their search parameters, that FragPipe provides significantly more identified proteins and peptides compared to MaxQuant.  This possibility is also supported by the incongruent result of low number of protein identifications versus adequate signal and MS/MS quality in raw data files.

-Related to the previous point, have the authors explored the possibility of proteins from other microorganisms/viruses?  The samples are a rich resource and these possibilities should be explored with the latest tools and sequence databases.

Minor issues

The use of silver stained gels is useful for qualitative comparisons, especially detection of molecular weight bias, but quantitative comparisons are challenging. Different proteins differentially bind silver ions, so if extracts have different protein compositions, their staining intensities may be different. The authors stated in the methods that a typical A205 measurement with a nanodrop was used, which should give more confidence in protein yield between different acid extraction and protein precipitation strategies. If possible, the authors can report protein yields in a table from this method.

A table(s) of protein IDs and their associated peptide assignments should be provided as supplemental information for modern (1 and 2 hr gradients) and archaeological specimens

Author Response

Dear reviewer, please find our answers to the comments attached.

Reviewer 2 Report

In their article Bender et al describe the optimization of a protocol applied  to extract proteins from dental calculus. Overall the article is well written and of interest to the scientists who work in that field. However,  I have the impression that some aspects have been treated superficially and that some information is lacking. In particular I refer to the determination of protein concentration. It is not clear why they did not use a typical method (Bradford/BCA/Lowry) to determine protein concentration. I understand from the article that they performed a spectrophotometric determination (at which wavelength, 205 nm, why ?) with the nanodrop  (line 250/251) device. It is well known that this device, while being very reliable for DNA/RNA determinations, has problems of reliability with proteins (check the literature). Moreover, the authors do not provide data. I did not find any indication throughout the text about this concentration. To be honest, I cannot believe that the manual crushing of calculus does not cause loss of material. I ask the authors to prepare a simple table containing these data at each step.

-line 233. ...mechanical sonication and sonication. Is not clear to me !

-line 288. the indication of the length of the alkyl chain (C8,C18 ??) is missing. The explanation of the gradient applied  is very confused.

- line 300. The description of the gels is a little "obsolescent". The "parts"have been always written as  stacking and running gels. Please check and change.

Author Response

(The authors gave the same response as above.)

Round 2

Reviewer 1 Report

The authors have addressed all comments. I look forward to their continued development of these valuable proteomic methods and application to diverse biological samples.

Reviewer 2 Report

The authors have replied to the reviewer's concerns and have slightly modified the text according to the requests. In my opinion the article may be accepted in its current form.